# Elucidating protonation pathways in $CO_2$ photoreduction using the kinetic isotope effect

Shikang Yin[1,2], Yiying Zhou[1], Zhonghuan Liu[1], Huijie Wang[1], Xiaoxue Zhao[1], Zhi Zhu[1], Yan Yan [1] ✉ & Pengwei Huo [1] ✉

The surge in anthropogenic $CO_2$ emissions from fossil fuel dependence demands innovative solutions, such as artificial photosynthesis, to convert $CO_2$ into value-added products. Unraveling the $CO_2$ photoreduction mechanism at the molecular level is vital for developing high-performance photocatalysts. Here we show kinetic isotope effect evidence for the contested protonation pathway for $CO_2$ photoreduction on $TiO_2$ nanoparticles, which challenges the long-held assumption of electron-initiated activation. Employing isotopically labeled $H_2O/D_2O$ and in-situ diffuse reflectance infrared Fourier transform spectroscopy, we observe $H^+/D^+$-protonated intermediates on $TiO_2$ nanoparticles and capture their inverse decay kinetic isotope effect. Our findings significantly broaden our understanding of the $CO_2$ uptake mechanism in semiconductor photocatalysts.

The continued dependence on fossil fuels has led to a substantial increase in anthropogenic carbon dioxide ($CO_2$) emissions, culminating in deleterious environmental impacts and energy crises[1,2]. An optimal strategy for addressing these challenges involves the conversion of $CO_2$ into value-added products, such as CO and $CH_4$, through artificial photosynthesis, which directly exploits incident sunlight and water[3,4]. However, a comprehensive understanding of the complex $CO_2$ photoreduction reaction at the molecular level, particularly at the $CO_2/H_2O$/catalyst gas-liquid-solid interface, remains elusive owing to the involvement of numerous proton-coupled electron transfer processes and potential reaction pathways with various intermediates[5–7]. Elucidating the $CO_2$ reduction pathway on the semiconductor catalyst surface is crucial for designing high-performance photocatalysts[8].

Upon light exposure, a comprehensive $CO_2$ photoreduction process typically encompasses water oxidation (or organic sacrificial agents, if utilized) and $CO_2$ reduction half-reactions. The water oxidation half-reaction is often regarded as analogous to the oxygen-evolving reaction (OER) in water-splitting[9,10]. The $CO_2$ reduction reaction encompasses multiple step-wise proton/electron transfer processes. Identifying the rate-determining step in such multi-step chemical reactions is an arduous task, yet essential for optimizing

reaction systems. For example, the classic $CO_2 + 2e^- + 2H^+ \rightarrow CO + H_2O$ (−0.53 V vs. NHE) reaction on a semiconductor photocatalyst necessitates the enrichment and activation of $CO_2$ molecules at the gas-vapor-catalyst or gas-liquid-catalyst interface, followed by a reduction reaction through a series of fundamental steps involving consecutive proton and electron transfers[11]. As a linear non-polar molecule, $CO_2$ is among the most stable carbon compounds. Nevertheless, the oxygen atoms in $CO_2$ can donate their lone pair of electrons to surface Lewis acid centers or be protonated by Brønsted acids[12]. The carbon atom can also accept electrons from Lewis base centers, forming carbonate-like species[13]. Moreover, the π electrons of the C=O bond can interact with electron centers, leading to bond cleavage and hybridization changes from O-$sp^2$ to O-$sp^3$. On the surface of the semiconductor catalyst, the adsorption configuration of $CO_2$ is also notably altered and influenced by the presence of water or other molecular proton donors[14–16]. All these potential reaction configurations constitute the initial steps of $CO_2$ activation.

Figure 1 illustrates two feasible reaction pathways for $CO_2$ photoreduction to CO in an aqueous solution: the electron-initiated pathway (path I) and the protonation pathway (path II). For a prolonged time, the initial step of $CO_2$ activation was presumed to occur

[1]Institute of Green Chemistry and Chemical Technology, School of Chemistry and Chemical Engineering, Jiangsu University, Zhenjiang 212013, PR China.
✉e-mail: dgy5212004@163.com; huopw@ujs.edu.cn

through path I, with a negatively charged $CO_2^{\delta\cdot-}$ species as the sole intermediate product[17,18]. However, the single-electron transfer to $CO_2$ is highly endergonic due to the molecule's negative adiabatic electron affinity[19]. Additionally, the initial $CO_2$ uptake on hydrophilic surfaces of $MO_x/MS_x$ semiconductor photocatalysts is challenging, which impedes direct single-electron transfer[20,21]. Instead, a protonation pathway (path II) that first polarizes $CO_2$ molecules, akin to the photocatalytic dehalogenation of non-polar halogenated aromatics[22], appears more plausible. However, both pathways lack definitive, direct evidence for confirmation.

The kinetic isotope effect (KIE) is a crucial and sensitive tool for investigating reaction mechanisms by tracking the transition state of the rate-determining step using isotopically-labeled reagents[23,24]. In this study, we employed isotopically labeled $H_2O/D_2O$ to determine an inverse kinetic solvent isotope effect (KSIE) of 0.2-0.9 on the photoreduction of $CO_2$ to CO on $TiO_2$ nanoparticles. Our findings confirm the protonation pathway with O $sp^2$–hybridized $O=C=O\cdot H^+/D^+$ intermediates (Fig. 2), providing the elucidation of the protonation pathway for $CO_2$ photoreduction and shedding light on the nature of $CO_2$ uptake on semiconductor photocatalysts.

**Path I:** $CO_2 \xrightarrow{e^-} \cdot CO_2^- \xrightarrow{H^+} \cdot COOH \xrightarrow[H_2O]{e^-/H^+} CO$

**Path II:** $CO_2 \xrightarrow{H^+} COOH^+ \xrightarrow{e^-} \cdot COOH \xrightarrow[H_2O]{e^-/H^+} CO$

**Fig. 1 | Initial $CO_2$ reduction mechanism.** Two feasible reaction pathways for photoreduction of $CO_2$ in aqueous solution.

## Results

### Inverse KIE of $CO_2$ photoreduction

We commenced our investigation by examining the KSIE of $CO_2$ photoreduction to CO in a $TiO_2$/water system, employing isotopically labeled $H_2O/D_2O$, and compared it to the water-splitting reaction in analogous systems (with or without $CO_2$). We first used the commercially available anatase $TiO_2$ (with ~20 nm-sized nanoparticles), a prevalent photocatalyst for water-splitting and $CO_2$ reduction, as a representative example of conventional metal oxide ($MO_x$) semiconductor catalysts with hydrophilic surfaces. We quantitatively detected the reduction products (i.e., $H_2$, $D_2$, CO) of the water-splitting and $CO_2$ photoreduction reactions through gas chromatography (Supplementary Fig. 1). Control experiments conducted without $CO_2$ yielded negligible amounts of CO, suggesting that $CO_2$ reduction primarily contributes to the product formation (Supplementary Fig. 2).

Figure 3a illustrates that $H_2$ production from the overall water-splitting in the $Pt\text{-}TiO_2/H_2O$ system (with Pt as the hydrogen evolution reaction (HER) cocatalyst) proceeds more swiftly than with $D_2O$, exhibiting a normal $KSIE_{H2O/D2O}(H_2)$ of 2.8 at 15 °C. Diminishing the reaction system's temperature augments the KSIE value to 5.8. The same experimental phenomena could be observed regardless of whether the cocatalyst was preloaded or loaded during the reaction (Supplementary Fig. 3). This temperature-dependent KSIE is consistent with the primary KIE's characteristics for O-H/O-D cleavage during the oxygen evolution reaction (OER), indicating direct O-H cleavage as the rate-determining step of water-splitting[25–27]. However, we observed an inverse $KSIE_{H2O/D2O}(CO)$ using the same catalyst in the presence of $CO_2$ (Fig. 3b). As the temperature declined from 15 to 3 °C, the $KSIE_{H2O/D2O}(CO)$ decreased from 0.9 to 0.2. Except for adding $CO_2$, all experimental conditions were congruent with the water-splitting reaction. Furthermore, the $KSIE_{H2O/D2O}(H_2)$ under identical experimental conditions displayed >1 normal values (Supplementary Fig. 4), suggesting different rate-determining steps between $CO_2$ photoreduction and water-splitting. Without Pt loading, the $CO_2$ photoreduction on

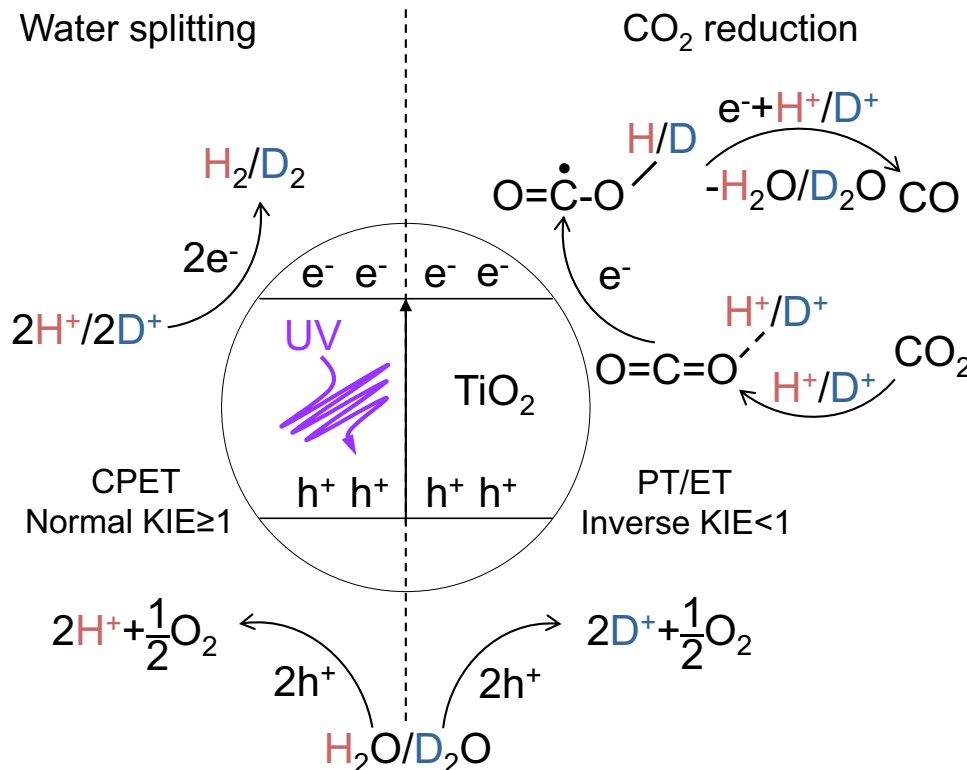

**Fig. 2 | Water-splitting and $CO_2$ photoreduction processes on $TiO_2$.** The water-splitting reaction (left) and the $CO_2$ photoreduction to CO (right) with isotopically labeled $H_2O/D_2O$. CPET represents concerted proton-coupled electron transfer, PT represents proton transfer, ET represents electron transfer.

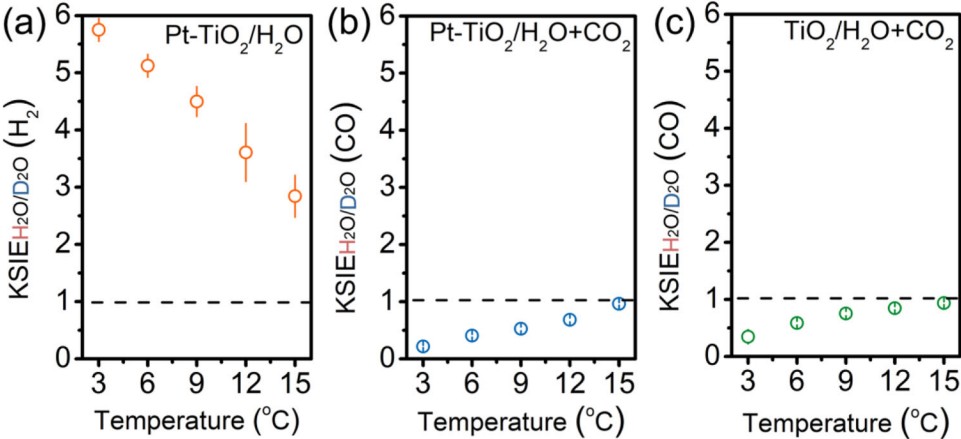

**Fig. 3 | Comparison of kinetic solvent isotope effect (KSIE) in different reaction systems. a** KSIE ($H_2$) values obtained by comparing the $H_2$ production kinetics of the water-splitting reaction on anatase $TiO_2$ in $H_2O/D_2O$ systems at different temperatures (Pt was loaded as cocatalysts, 3% chloroplatinic acid); **b** KSIE (CO) values obtained by comparing the kinetics of the $CO_2$ reduction reaction on anatase $TiO_2$ in $H_2O/D_2O$ systems at different temperatures (Pt was loaded as cocatalysts); **c** KSIE (CO) values are given by comparing the CO production kinetics of the $CO_2$ reduction reaction in the $H_2O/D_2O$ systems at different temperatures without Pt cocatalysts. Error bar represents three independent experiments obtaining the standard deviation.

pristine $TiO_2$ exhibited analogous inverse $KSIE_{H_2O/D_2O}$(CO) values (Fig. 3c). This outcome implies that the rate-determining step encompasses hybridization changes from $sp^2$ to $sp^3$ in the secondary inverse KIE phenomenon, consistent with the double-bond break of O=C=O molecules instead of direct O-H cleavage in OER. By employing $H_2O/D_2O$ as labeled isotopes, the observed inverse KIE denotes a configuration transition between protonated intermediates O=C=O-$H^+/D^+$ (O $sp^2$) and O=$C^-$-O-H/D (O $sp^3$) during electron transfer, offering robust evidence for a protonation pathway involving the formation of the protonated intermediate O=C=O-$H^+$ as the initial step of $CO_2$ photoreduction (path II, Fig. 1). This mechanism challenges the widely accepted electron-initiated pathway (path I, Fig. 1). Note that such a protonation pathway does not rely on the presence of a water solvent. We introduced water in the form of vapor into the reaction instead of as a solvent, and the same inverse KIE could be observed (Supplementary Fig. 5). This suggests that the protonation of $CO_2$ can be achieved through water vapor.

It is well-acknowledged that the characteristics of employed $TiO_2$ catalysts can significantly influence their interaction with target molecules and thus lead to the change in reaction kinetics. To better ascertain whether the observed KIE changes originated from the reaction pathway itself or were influenced by the catalyst material, we conducted supplementary experiments across various $TiO_2$ systems to bolster our findings. We first examined the influence of the $TiO_2$ crystal structure by comparing the KIE for the $CO_2$ reduction on anatase and rutile $TiO_2$ (characterized by XRD and TEM/HR-TEM, see Supplementary Fig. 6 and Supplementary Fig. 7). We found that the $KSIE_{H_2O/D_2O}$ (CO) on both anatase and rutile catalysts exhibited inverse KIE (<1), suggesting that the observed inverse KIE and the protonation pathway in $CO_2$ reduction are common to both crystal structures. Furthermore, we examined the effect of exposed facets of the $TiO_2$ catalyst. As a comparison, we synthesized anatase $TiO_2$ nanosheet with high exposure of the {001} facet according to a reported method[28], which was characterized using XRD, TEM, HR-TEM, and SAED (Supplementary Fig. 8). The $KSIE_{H_2O/D_2O}$ (CO) for $CO_2$ reduction on these {001}-exposed $TiO_2$ nanosheets still exhibited inverse KIE < 1. These results further confirmed that the exposed facet of the $TiO_2$ nanoparticles does not influence the $CO_2$ reduction pathway under our experimental conditions. Finally, we evaluated the effect of oxygen defects. Oxygen vacancies on the $TiO_2$ surface are often considered active sites for the oxygen evolution reaction (OER)[29]. However, their direct influence on $CO_2$ reduction is less clear. We prepared oxygen-deficient $TiO_2$

nanoparticles according to a reported method of $NaBH_4$ calcination[30], and characterized them using XRD, TEM, and ESR, which confirmed the presence of oxygen vacancies (Supplementary Fig. 9). The $KSIE_{H_2O/D_2O}$ (CO) for $CO_2$ reduction on these oxygen-deficient nanoparticles remained <1, exhibiting the secondary inverse KIE and aligning with the protonation pathway. These additional characterizations and experiments confirmed that the inverse KIE observed in the $CO_2$ reduction reaction is intrinsic to the $TiO_2$ material generalized to a broader range of $TiO_2$-based photocatalytic systems, regardless of the crystal structure, exposed facet, or oxygen vacancy concentration.

As a conventional metal oxide semiconductor with a hydrophilic surface and moderate reduction ability, $TiO_2$ demonstrates inadequate $CO_2$ uptake capacity[31]. As a result, the endergonic single-electron transfer of $CO_2 \rightarrow CO_2^{\delta-}$ on $TiO_2$ signifies a high-energy reaction. Nevertheless, in the photocatalytic dehalogenation of non-polar halogenated aromatics (e.g., polybrominated diphenyl ethers, PBDEs), a protonation pathway involving initial proton adhesion on the aromatic ring of PBDE molecules before subsequent electron transfer has been substantiated[22]. Additionally, in our recent work, we uncovered a step-wise proton transfer/electron transfer (PT/ET) pathway on $TiO_2$ for the single-electron/single-proton reduction of $^tBu_3ArO\bullet$ and TEMPO$\bullet$ to $^tBu_3ArOH$ and TEMPOH[32]. These investigations support the feasibility of the protonation pathway for $CO_2$ photoreduction on $TiO_2$ catalysts.

## In-situ DRIFTS measurements

To investigate the protonation pathway and monitor O=C=O-$H^+/D^+$ intermediates during photocatalytic $CO_2$ reduction, we employed in-situ diffuse reflection infrared Fourier transform spectroscopy (DRIFTS) at the $TiO_2/H_2O/CO_2$ ($TiO_2/D_2O/CO_2$) interface. The experiment was carried out under 365 nm irradiation (3 W, LED) for 15 min, with $H_2O$ and $CO_2$ (5 mL/min) introduced into the chamber by $N_2$ flow (5 mL/min) until equilibrium was reached. We used the pre-reaction equilibrium system as a blank background and observed negative or positive IR signals during the reaction, indicating the loss or gain of species at the $TiO_2/H_2O/CO_2$ ($TiO_2/D_2O/CO_2$) interface. Control experiments demonstrated that in the absence of incident light, the reaction did not occur (Supplementary Fig. 10).

Figure 4a reveals negative peaks at 3700–2800 $cm^{-1}$ and 1665 $cm^{-1}$ at the $TiO_2/H_2O$ interface upon constant irradiation, corresponding to the O-H stretching and H-O-H bending vibrations of $H_2O$ molecules[33], respectively. The weak signal at 3705 $cm^{-1}$ represented the terminal

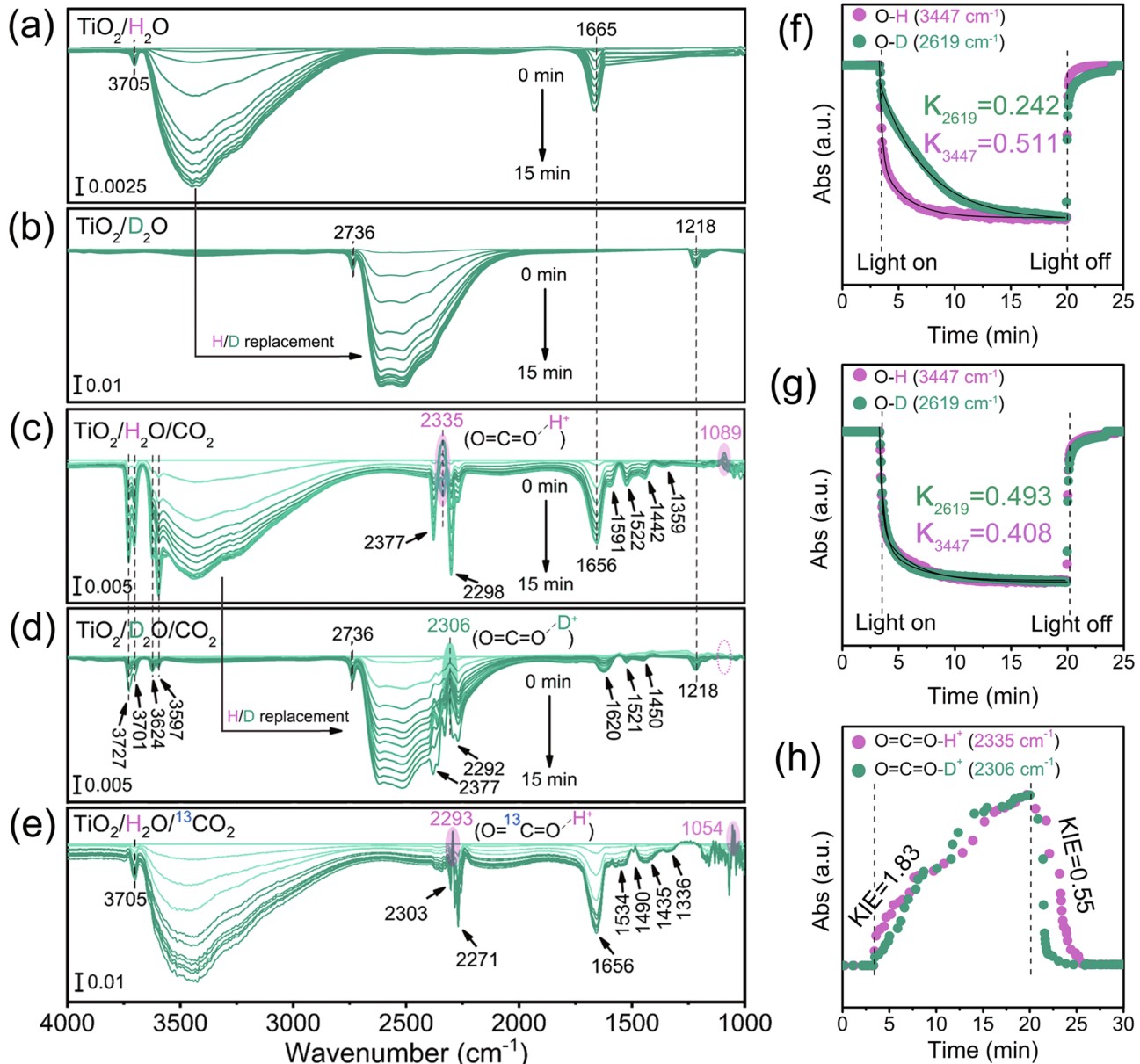

**Fig. 4 | In-situ diffuse reflectance infrared Fourier transform spectroscopy (DRIFTS) measurements.** DRIFTS spectra collected at the $TiO_2$/$H_2O$ (a), $TiO_2$/$D_2O$ (b), $TiO_2$/$H_2O$/$CO_2$ (c), $TiO_2$/$D_2O$/$CO_2$ (d) and $TiO_2$/$H_2O$/$^{13}CO_2$ (e) interfaces under constant 365 nm (3 W, LED) irradiation in 15 min; **f** Time profiles of IR signals at 3447 $cm^{-1}$ in (a) and 2619 $cm^{-1}$ in (b) from light-on to light-off, representing the decay kinetics of O-H and O-D in water-splitting; **g** Time profiles of IR signals at 3447 $cm^{-1}$ in (c) and 2619 $cm^{-1}$ in (d) from light-on to light-off, representing the

decay kinetics of O-H and O-D in $CO_2$ photoreduction; **h** Time profiles of IR signals at 2335 $cm^{-1}$ and 2306 $cm^{-1}$ (after baseline corrections to maintain positive values) from light-on to light-off, representing the formation and decay kinetics of O=C=O-$H^+$ and O=C=O-$D^+$. An inverse KIE was obtained during the decay process after light-off. Pink shading represents the peak position of the COOH$^+$ intermediate, and green shading represents the peak position of the COOD$^+$ intermediate.

O-H group on the $TiO_2$ surface[34]. When $H_2O$ was replaced with $D_2O$, noticeable redshifts of both O-D stretching and D-O-D bending vibrations to 2700–2100 $cm^{-1}$ and 1218 $cm^{-1}$ were observed (Fig. 4b), in line with the theoretical H/D replacement effect[35,36]. The decay kinetics of O-H/O-D stretching vibrations showed that the O-H signal decays much faster than the O-D signal, resulting in a direct KIE of 2.11 (Fig. 4e), consistent with the measured normal KSIE$_{H2O/D2O}$($H_2$) values and representing features of the direct O-H/-D cleavage during overall water-splitting.

In the $TiO_2$/$H_2O$/$CO_2$ system (Fig. 4c), negative peaks at 2377 $cm^{-1}$ and 2298 $cm^{-1}$ corresponding to the C=O stretching vibrations of $CO_2$ were observed, along with an emerging positive signal peak at 2335 $cm^{-1}$ adjacent to the decayed stretching vibration signals of $CO_2$,

likely due to the formation of the protonated $CO_2$ intermediate (O=C=O-$H^+$). The adhesion of a proton to the oxygen atom would alter the C=O bond and alter the effective mass of oxygen, thereby changing the vibration frequency. In addition, according to Hooke's Law, the adhesion of protons to the oxygen nucleus in C=O bonds increases the effective mass of the oxygen atom, which subsequently results in a change in the frequency of the stretching vibrations of the C=O bond[37]. Moreover, the increasing positive signal at 1089 $cm^{-1}$ is likely from the C=O-$H^+$ bending vibration. Negative peaks at 3727 $cm^{-1}$, 3701 $cm^{-1}$, 3624 $cm^{-1}$, and 3597 $cm^{-1}$ corresponded to the weak overtone region of $CO_2$ molecules[38], and signals at 1591 $cm^{-1}$, 1522 $cm^{-1}$, 1442 $cm^{-1}$, and 1359 $cm^{-1}$ were assigned to -COOH* species, monodentate carbonate (m-$CO_{2-3}$) groups, as well as the antisymmetric and symmetric

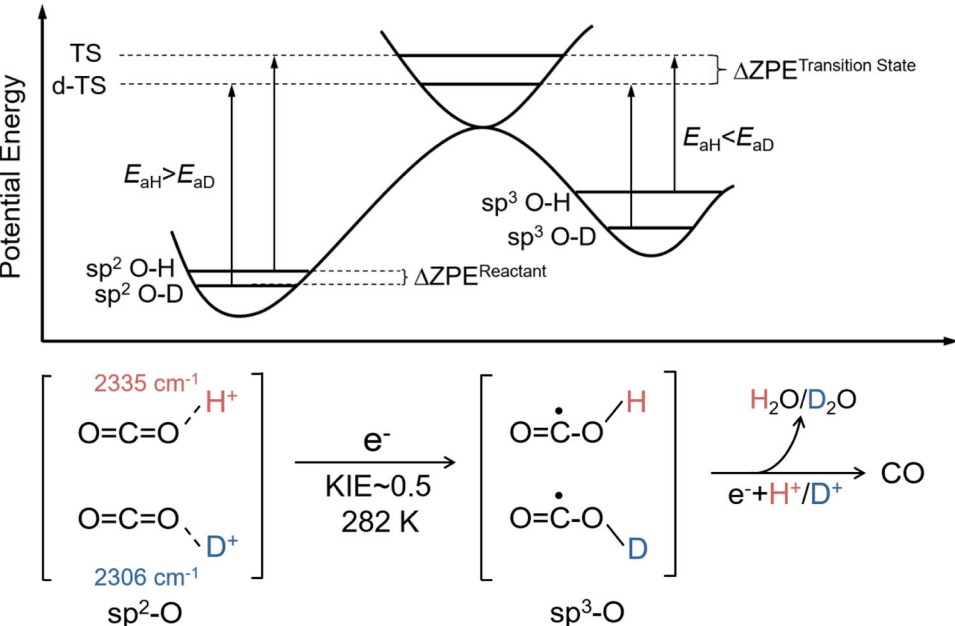

**Fig. 5 | The source of inverse KIE.** Schematic illustrations and energetic profiles of the O=C=O·H$^+$/D$^+$ → O=C$^-$–O·H/D electron transfer process. TS represents transition state, ZPE represents zero-point energy.

stretching bands of bidentate carbonate (b-CO$_{2-3}$) groups[39,40], respectively.

To verify that the observed changes in CO$_2$ FT-IR signals resulted from a surface reaction rather than a modification in the surface adsorption configuration of CO$_2$ under incident light, we carried out a control experiment. This involved first running the reaction for a specified time under light, followed by the removal of the gas phase using a N$_2$ flow. By subtracting the equilibrium background in N$_2$ prior to the experiment, we were able to observe changes in surface-adsorbed species over time. Given that the removal of the CO$_2$ gas phase would cut off the replenishment of surface CO$_2$, a fading CO$_2$ signal would suggest that the observed signals stemmed from the reaction rather than adsorption. Otherwise, we would observe unchanged, stable adsorbate signals. As illustrated in Supplementary Fig. 11c, d, after the abrupt removal of CO$_2$, both the negative and positive signals of C=O vibration from CO$_2$ species around 2330 cm$^{-1}$ to 2340 cm$^{-1}$ continued to decrease over time and vanished within tens of seconds. This suggests that the observed CO$_2$ signals are not from a stable adsorbate but from a surface reaction. Furthermore, to validate the assignment of the protonated O=C=O·H$^+$ intermediate, we replaced H$_2$O with deuterated-labeled D$_2$O under identical conditions. The diagnostic signal peak of the protonated intermediate shifted towards a lower wavenumber from 2335 cm$^{-1}$ to 2306 cm$^{-1}$ upon replacing O=C=O·H$^+$ with O=C=O·D$^+$ (Fig. 4d). The negative signal peaks (both stretching bands and overtone region) of CO$_2$ molecules remained unchanged. This H/D replacement effect on the C=O stretching vibration of O=C=O·H$^+$/D$^+$ intermediates is consistent with the results of Hooke's Law (detailed calculation formula see supplementary methods). However, the C=O·D$^+$ bending vibration was not observed in O=C=O·D$^+$, which likely shifts a lower frequency, beyond our in-situ DRIFTS detection range (Fig. 4d). Together with the H/D replacement experiments without CO$_2$, the shift of the diagnostic peak of O=C=O·D$^+$ compared to that of the unlabeled O=C=O·H$^+$ provides direct evidence for the formation of protonated O=C=O·H$^+$ intermediates during the CO$_2$ photoreduction process at the TiO$_2$/H$_2$O/CO$_2$ interface. Furthermore, we have an additional DRIFTS experiment using $^{13}$C-labeled $^{13}$CO$_2$. As depicted in Fig. 4e, a distinct redshift from 2335 cm$^{-1}$ to 2293 cm$^{-1}$ of the C=O stretching vibration was observed when employing $^{13}$CO$_2$, corresponding to the shift of the $^{13}$C=O

stretching vibration signal in O=$^{13}$C=O·H$^+$ compared to the unlabeled $^{12}$C=O in O=C=O·H$^+$/D$^+$ (2335 cm$^{-1}$/2306 cm$^{-1}$) due to the $^{12}$C/$^{13}$C isotope replacement effect. Moreover, the bending vibration of C=O·H$^+$ at 1089 cm$^{-1}$ was also shifted to 1054 cm$^{-1}$ in the $^{13}$CO$_2$ system corresponding to $^{13}$C=O·H$^+$. These findings are highly consistent with our KIE experimental results and further validates our assignment.

## Quantum chemical calculations

We further conducted quantum chemical calculations to simulate the infrared signals of the H$^+$/D$^+$ protons adhered to the oxygen atom in CO$_2$. The results are consistent with our assumption that the C=O stretching vibration in CO$_2$ does not form a C-O-H sp$^3$ structure after adhering to a H$^+$/D$^+$ proton, thereby a C-O signal does not appear (Supplementary Fig. 12). It remains at 2300–2400 cm$^{-1}$ (the discrepancy between the calculation and actual data should come from different adsorption interfaces; the calculation only simulates the situation in a vacuum). The vibration frequency changes from protonated species and pristine CO$_2$ due to the influence of bond energy and the effective mass of oxygen. Moreover, replacing H$^+$ with D$^+$ indeed causes the simulated C=O stretching vibration to shift to a lower frequency (2403 cm$^{-1}$ → 2394 cm$^{-1}$). Interestingly, quantum calculations also reveal possible O-H/O-D stretching vibrations (3406 cm$^{-1}$/2490 cm$^{-1}$), which are not clearly observed in the actual experiment due to the significant influence of water signals. More importantly, we found that the 960 cm$^{-1}$ in O=C=O·H$^+$ corresponds to the bending vibration of C=O·H$^+$, which correspond to the positive signal at 1089 cm$^{-1}$ observed in in-situ DRIFTS. In O=C=O·D$^+$, the bending vibration of C=O·D$^+$ shifts to a lower frequency, beyond our in-situ DRIFTS detection range, fully consistent with our observation. However, when $^{13}$C is used for simulation, the bending vibration of $^{13}$C=O·H$^+$ can be seen to shift from 960 cm$^{-1}$ to 952 cm$^{-1}$. In our actual in-situ DRIFTS, when using $^{13}$CO$_2$, we indeed observed a shift towards a lower wavenumber of the $^{13}$C=O·H+ bending vibration (1054 cm$^{-1}$) from C=O·H$^+$ (1089 cm$^{-1}$) using unlabeled CO$_2$ (Fig. 4e). This result fully support our assignment of the O=C=O·H$^+$ signal.

## Discussion

In prior research, the generation of CO$_2$$^{δ-}$ anion radicals have been detected during the photocatalytic degradation of formate on TiO$_2$

nanoparticles using infrared (IR) and electron spin resonance (ESR) spectroscopy[41,42]. Although the $CO_2^{\delta-\bullet}$ anion radical is often cited as the exclusive intermediate of the initial step in $CO_2$ photoreduction, no definitive evidence has been provided for its presence in $CO_2$ photoreduction systems. On a polar $TiO_2$ surface surrounded by $H_2O$ molecules, chemisorbed species, mainly $OH^-$, produce distinct π or δ resonances, while physisorbed species have weak signals[43]. This limits the opportunities for single-electron transfer by neutral physisorbed $CO_2$ molecules, which are scarce at the polar $H_2O/TiO_2$ interface. Instead, an ionized $CO_2$ moiety promotes interfacial $CO_2$ uptake[44], facilitating subsequent electron/proton transfer. Under our experimental conditions, the only visible positive signal peak after light illumination corresponds to the protonated O=C=O·H$^+$/D$^+$ signal. This finding contradicts previous understandings of the $CO_2$ photoreduction mechanism and suggests a protonation pathway[45].

We also compared the decay kinetics of O=C=O·H$^+$/D$^+$ and O-H/O-D during the reaction. The inverse kinetic isotope effect (KIE) of the O sp$^2$ → O sp$^3$ hybrid transition process is a classic phenomenon in reaction kinetics associated with the disparity in the vibration frequency of chemical bonds[22,46]. The observed inverse KIE of O=C=O·H$^+$/D$^+$ decay (KIE = 0.55) provides strong evidence for the protonation pathway (Fig. 4h; Supplementary Fig. 13), which involves the O=C=O·H$^+$/D$^+$ → O=C$^-$-O·H/D double-bond break with a hybridization change from O sp$^2$ → O sp$^3$ via additional electron transfer (Fig. 5). In most reactions, the overall rate is determined by the slowest step, known as the rate-determining step[47]. In our system, without $CO_2$, the direct breakage of the O-H/O-D bond undeniably constitutes the rate-determining step, hence its KIE is greater than 1 (Fig. 4f). However, the decay kinetics of O-H/O-D stretching vibration also exhibited an inverse KIE = 0.827 in the presence of $CO_2$ (Fig. 4g), indicating that the slower reduction reaction of $CO_2$ (in this case, the reduction of the protonated intermediate) becomes the rate-determining step.

In this study, we unveil a mechanism governing the photoreduction of $CO_2$ on semiconductor catalysts, which transpires via a protonation pathway. We report the formation of an O=C=O·H$^+$ intermediate, which exhibits an inverse KIE during the subsequent electron transfer process. This electron transfer process prompts the conversion of the sp$^2$−hybridized O=C=O·H$^+$/D$^+$ species into the sp$^3$−hybridized O=C$^-$-O·H/D species. Utilizing isotopically labeled in-situ DRIFTS, we successfully discern the formation of H$^+$/D$^+$-protonated O=C=O·H$^+$/D$^+$ intermediates on $TiO_2$ nanoparticles and capture their inverse decay KIE. This research substantially broadens our comprehension of the $CO_2$ uptake mechanism in semiconductor photocatalysts, necessitating a re-examination of long-held assumptions within the field. Our findings hold significant potential for advancing the development of more efficient and sustainable photocatalytic $CO_2$ reduction technologies in the future.

## Methods
### Materials
Commercial titanium dioxide ($TiO_2$, anatase, 20 nm), sodium borohydride ($NaBH_4$), tetrabutyl titanate, hydrofluoric acid (HF, 40 wt%), chloroplatinic acid ($H_2PtCl_6\cdot6H_2O$), ethanol and deuterium oxide ($D_2O$, 99.9 atom % D) were purchased from Shanghai McLean Biochemical Technology Co., Ltd. All reagents used in the synthesis were analytically pure and had not been further purified. Deionized water was obtained from a purified distillation unit in the laboratory. Before any photocatalytic reaction experiments, $TiO_2$ samples were first calcinated and then illuminated by an ultraviolet lamp (365 nm, 160 mW·cm$^{-2}$) in water.

### Synthesis of (001) exposed $TiO_2$ nanosheet
In a typical synthesis, 12.5 mL of tetrabutyl titanate was mixed with 2 mL of HF solution, under stirring for 30 min. The solution was then transferred into a 50-mL Teflon-lined autoclave, and kept at 180 °C for

24 h. After the solvothermal reaction, the resulting white precipitates were collected and washed with ethanol and distilled water for three times. The samples were dried in a vacuum oven at 60 °C for 12 h.

### Synthesis of oxygen-deficient $TiO_2$
1 g $TiO_2$ nanoparticle powder was mixed with 2 g $NaBH_4$ and the mixture was ground for 30 min thoroughly. Then the mixture was transferred into a porcelain boat, and placed in a tubular furnace, heated from room temperature to 350 °C/1 h under an Ar atmosphere at a heating rate of 10 °C min$^{-1}$. After naturally cooling down to room temperature, the colored $TiO_2$ was obtained, simply washed with deionized water and ethanol several times to remove unreacted $NaBH_4$, and dried at 70 °C.

### Water-splitting experiments
In a typical procedure, 50 mg $TiO_2$ powder was dispersed in 10 mL deionized water ($H_2O$) and 10 mL deuterium water ($D_2O$), respectively. Next, 3 wt% Pt as cocatalysts was loaded via in-situ photo deposition using $H_2PtCl_6\cdot6H_2O$ without any sacrificial agents. After irradiation with an ultraviolet lamp (365 nm, 160 mW·cm$^{-2}$), Gas products were determined by using a gas chromatography (GC-7900) equipped with the TCD thermal conductivity detector and the carrier gas was chosen Ar.

### $CO_2$ photoreduction experiments
$CO_2$ photoreduction was carried out in a sealed self-made 150 mL stainless-steel reactor with an ultraviolet lamp (365 nm, 160 mW·cm$^{-2}$) as the light source. In a typical procedure, 50 mg catalyst was dispersed in 10 mL deionized water ($H_2O$) and 10 mL deuterium water ($D_2O$), respectively. $CO_2$ was then introduced into the reactor and bubbled for 25 min to completely remove air. Gas products were detected by the gas chromatography (GC-7920, China) equipped with hydrogen flame ionization detector (FID) and thermal conductivity detector (TCD). In addition, the control experiment had the same experimental conditions as described above except for the addition of 3 wt% Pt as cocatalysts; In the gas-solid reaction system, 50 mg catalyst was dispersed in quartz grooves, add 2 ml of water or deuterated water to the bottom of the 150 ml reactor with no direct contact with the catalyst, assuring that water participates in the reaction in vapor state. $CO_2$ flow was then introduced into the reactor for 25 min before light-on.

### In-situ DRIFTS experiments
In-situ diffuse reflection infrared Fourier transform spectroscopy (DRIFTS) experiments were conducted on a Nicolet iS10 (Thermo) machine according to our previous work[47]. In a typical procedure, catalyst sample was sealed in the reaction chamber with a quartz window. $CO_2$ and $H_2O$ (or $D_2O$) were carried into the reaction chamber by $N_2$ flow until equilibrium. After taking the equilibrium system before reaction as the blank background, IR signals were collected in-situ during the incident irradiation of a 365 nm LED lamb (3 W) through the quartz glass window.

### Hooke's law
Taking diatomic as an example, when the diatomic is telescopic and vibrating, they can be approximated as a simple harmonic oscillator. Given two bodies, one with mass $m_1$ and the other with mass $m_2$, the equivalent one-body problem, with the position of one body with respect to the other as the unknown, is that of a single body of mass; where the equivalent mass of O=C=O·H$^+$ is $m_1$ = 12 (C), $m_2$ = 17 (O-H, $\nu_1$ = 2335 cm$^{-1}$); The equivalent mass of O=C=O·D$^+$ is $m_1$ = 12 (C), $m_2$ = 18 (O-D, $\nu_2$ = 2306 cm$^{-1}$).

$$\text{Composite mass}: \mu = \frac{1}{\frac{1}{m_1} + \frac{1}{m_2}} = \frac{m_1 m_2}{m_1 + m_2} \qquad (1)$$

$$\text{Vibration frequency} : \upsilon = \frac{1}{2\pi}\sqrt{\frac{k}{\mu}} \qquad (2)$$

When $\upsilon_1 = 2335\ cm^{-1}$:

The force constants of chemical bonds:

$$k = \mu(2\pi\upsilon_1)^2 = \frac{m_1 m_2}{m_1 + m_2}(2\pi\upsilon_1)^2 = \frac{12 \times 17}{12 + 17}(2 \times 3.14 \times 2335)^2 = 1.51 \times 10^9$$

When the equivalent mass of $O=C=O\cdot D^+$ is $m_1 = 12$, $m_2 = 18$:

$$\upsilon_2 = \frac{1}{2\pi}\sqrt{\frac{k}{\mu}} = \frac{1}{2\pi}\sqrt{\frac{k}{\frac{m_1 m_2}{m_1 + m_2}}} = \frac{1}{2 \times 3.14}\sqrt{\frac{1.51 \times 10^9}{\frac{12 \times 18}{12 + 18}}} = 2308$$

## Data availability

The data supporting the findings of this study are available within the article and its Supplementary Information files. All other relevant source data are available from the corresponding author upon request. Source data are provided with this paper.

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

## Acknowledgements

We gratefully acknowledge the financial support of the National Natural Science Foundation of China (Grant No. 22078131 (P. Huo) and 22208127 (Z. Zhu)); The Science and Technology Planning Social Development Project of Zhenjiang City (SH2021013 (P. Huo)); Graduate Research and Innovation Projects of Jiangsu Province (Grant No. KYCX22_3696 (S. Yin)).

## Author contributions

S.Y. and Y.Y. designed the whole experiment. S.Y., Y.Z., Z.L., H.W., and X.Z. conducted most experiments. S.Y. and Y.Y. wrote the paper. Z.Z. and P.H. contributed to the data analysis of the paper quality through discussions.

## Competing interests

The authors declare no competing interests.
