## [Peer Review File · Nature Communications]

Elucidating protonation pathways in CO₂ photoreduction using the kinetic isotope effectREVIEWER COMMENTS

Reviewer #1 (Remarks to the Author):

This study is concerned with an investigation regarding the initial steps within the mechanism of photocatalytic carbon dioxide reduction. Employing isotopic labelling of the solvent water the authors are able to show that apparently the carbon dioxide molecule is first protonated on the photocatalyst surface before its photocatalytic one-electron reduction takes place. This is a nice example for a proton coupled electron transfer mechanism and will be of interest to a wide community of researchers. Hence, the acceptance of this paper in this journal is suggested following some minor revision as indicated:

1) The importance of isotopic studies to understand reaction mechanisms in photocatalysis has recently been reviewed. This review should be cited and discussed in the introduction of the paper: C. Günnemann, D. W. Bahnemann, P. K. J. Robertson, "Isotope Effects in Photocatalysis: An Underexplored Issue", ACS Omega 6 (2021) 11113-11121

2) The usage of the English language could be improved.

Reviewer #2 (Remarks to the Author):

This is an original and a significant study on exploring the mechanism of the photocatalytic CO₂ reduction at the TiO₂ surface. The authors have utilized kinetic isotope effect to demonstrate the initial protonation step followed by the electron transfer in the PT-ET CO₂ photoreduction. The utilized mythology was well described and fulfill the required standards. However, the conclusions and data discussion needs to be improved. Hence, I would recommend the publication of this study after the followed comments have been addressed.

- The poor characteristics of the TiO₂ sample low drastically the impact of the study. It is well known that the interaction between the molecule and the semiconductor surface determines the photocatalytic mechanism. In case of TiO₂ the interactions vary depending on the polymorph, exposed facet, concentration of the oxygen vacancies and other defects, the presence of the hydroxyl groups, the Lewis and Bronstedt acidity, energetic position of the conduction band and valence band etc. The TiO₂ sample studied here, was not characterized, so the reported mechanism is limited to this particular sample and can not be generalized to other systems.

- Have the authors tried to perform DRIFTS experiments with labeled C¹³O₂ to rule out the electron transfer step occurring first.

- Control-DRIFTS-experiments of bare TiO₂, of TiO₂/H₂O, TiO₂/D₂O, TiO₂/H₂O/CO₂ and TiO₂/D₂O/CO₂ in the dark are missing.

- Page 9. Could the authors please explain this sentence: The decay kinetics of O-H/O-D stretching vibration also exhibited an inverse KIE=0.827 in the presence of CO₂ (Figure 2f), suggesting a changed rate-determining step

- Following relevant references are missing: ACS Catal. 2018, 8, 2, 1009–1017, Science 336, 1298 (2012).

- Page 6, Ref. 19 is not adequate.
- References are missing: Page 5, ...This temperature-dependent KSIE is consistent with the primary KIE's characteristics for O-H/O-D cleavage during the oxygen evolution reaction (OER), indicating direct O-H cleavage as the rate-determining step of water-splitting; Page 6: ...in line with the theoretical H/D replacement effect

Reviewer #3 (Remarks to the Author):

In their submitted article, Yan et al. attempt to study the kinetic isotope effect for Pt/TiO₂ and TiO₂ in CO₂ reduction and hydrogen evolution. Very different KIEs are observed in the different cases, which is attributed to different reaction intermediates. In particular, the authors propose the formation of an O=C=O-H intermediate as initial step in photocatalytic CO₂ reduction.

The attempt of the authors is well noted, and finding an answer to this intriguing question is of high relevance for the community. However, in the current state of the article, the authors do not achieve this goal. Too many questions are left unanswered, and the hypotheses are not solidly proven. It appears to be too much corrections for a major revision, so the article should be rejected in the current state. However, it might be possible that a completely rewritten version might be submitted again to Nature Communications in the future. In the following I will outline the various issues with the current version of the article.

(1) Minor issue: In the introduction, the presented theories on the adsorption of CO₂ and its photocatalytic activation on TiO₂ have also been published (early on) in some noteworthy articles from Europe and the USA. It is suggested to cite, in particular, the following articles: (i) M.A. Roberts, H.-J. Freund, *Surface Science Reports* 25 (1996) 225; (ii) I.A. Shkrob et al. *J. Phys. Chem. C* 116 (2012) 9461; (iii) A. Pougín et al. *Phys. Chem. Chem. Phys.* 18 (2016) 10809.

(2) Minor issue: Page 4: The term "polar surface" is not well chosen. Some semiconductors, for example ZnO, have actual polar surfaces, with a dipole moment perpendicular to the surface plane (see here: <https://doi.org/10.1021/acs.chemmater.7b01487>). The crystal structure of TiO₂ does not allow the existence of such polar surfaces by this definition. Instead, the authors might think of a better term, such as "hydrophilic surface" or "hydroxylated surface" or "favoring the adsorption of polar molecules" or other.

(3) Minor issue: The raw data of CO production on Pt/TiO₂ in presence of CO₂ should also be reported in Figure S1.

(4) Major issue: Some details on the experiments are not clear: Pt was deposited before the start of the hydrogen evolution experiment, but this would imply that the chloride from H₂PtCl₆ is still left in the solution. For the CO₂ reduction tests, it seems that Pt was already present beforehand, but then the

chloride would not be present. Could this also be the reason for the very different KSIE observed in the two cases?

(5) Major issue: In a related manner as in point 4 above, there is a significant discrepancy between the photocatalytic experiments (in liquid water) and the DRIFTS experiments (with water dosed from the gas phase). Can the water still be considered a solvent in the latter case? Are the reaction conditions sufficiently comparable?

(6) Major issue: Deducing the formation of an O=C=O-H intermediate from the given IR spectra is a very vague hypothesis. It is more likely that the light irradiation changes the charge distribution at the interface, making the adsorption of water less feasible and the CO₂ adsorption more feasible. The species with a vibrational band at 2335 cm⁻¹ might simply be weakly coordinated/bound/adsorbed CO₂ at the surface, which is also likely given the only minor shift compared to gaseous CO₂ (~2360 cm⁻¹). The "band" at 2306 cm⁻¹ is a mere guess, since it might alternatively be a dip between two other negative bands. At the same time, there is still a "band" (of similar size as the one assigned by the authors) at or near 2335 cm⁻¹, as before. I admit that DRIFTS is usually a vague technique, and there are not many alternatives (if any) to see adsorbates on real catalysts under reaction conditions. But maybe the authors can make a more clear band assignment if they perform another control experiment where they run the reaction for a certain time under light, then remove the gas phase, and look for stable adsorbates left after reaction. But of course, other methods to prove the suggested surface intermediates are also very welcome.

(7) Major issue: In continuation of comment 6, I would also suggest quantum chemical calculations of the vibrations of the O=C=O-H intermediate. The strong band of (gaseous or weakly adsorbed) CO₂ at/near 2360 cm⁻¹ is the asymmetric O=C=O stretching frequency. But in CO₂, both O atoms are identical, giving rise to only one band. A species in which one oxygen atom binds or coordinates a hydrogen atom would resemble more a carboxyl species, which features very different bands for the two different O=C and C-O(-H) bonds. I am doubtful that such a species would have one vibrational band near 2335 cm⁻¹.

(8) Minor issue: Page 7: "Hooker's law" should be "Hooke's law".

(9) Major issue: In the experimental section, the authors do not report any particular cleaning of the TiO₂ before the reaction. Formation of carbon-containing "products" from impurities is a known observation for TiO₂ (see works by Guido Mul, Elena Selli, Adriana Zaleska and Jennifer Strunk from ~2010 to 2019). Since the authors perform water splitting before adding CO₂, which can reasonably be considered a blank experiment for CO₂ reduction (see, e.g., Moustakas and Strunk, Chem. Eur. J. 24, 2018) such formation of byproducts should be observed. Did the authors really observe "no" CO (or CH₄) during the initial water splitting experiments? Or in other words, can the formation of CO from sources other than CO₂ be clearly excluded?

Reply to reviewers' comments

To Reviewer 1:

Comments:

This study is concerned with an investigation regarding the initial steps within the mechanism of photocatalytic carbon dioxide reduction. Employing isotopic labelling of the solvent water the authors are able to show that apparently the carbon dioxide molecule is first protonated on the photocatalyst surface before its photocatalytic one-electron reduction takes place. This is a nice example for a proton coupled electron transfer mechanism and will be of interest to a wide community of researchers. Hence, the acceptance of this paper in this journal is suggested following some minor revision as indicated:

Q1: *The importance of isotopic studies to understand reaction mechanisms in photocatalysis has recently been reviewed. This review should be cited and discussed in the introduction of the paper: C. Günnemann, D. W. Bahnemann, P. K. J. Robertson, "Isotope Effects in Photocatalysis: An Underexplored Issue", ACS Omega 6 (2021) 11113-11121.*

Response: We appreciate the reviewer's useful suggestion. The corresponding paper has been introduced and cited in the revised manuscript. Please see **Page 4, Line 10-11**.

Q2: *The usage of the English language could be improved*

Response: We appreciate the reviewer's comment. We carefully checked the English language in the revised manuscript as requested, which has substantially improved the readability of our work.

To Reviewer 2:

Comments:

This is an original and a significant study on exploring the mechanism of the photocatalytic CO₂ reduction at the TiO₂ surface. The authors have utilized kinetic isotope effect to demonstrate the initial protonation step followed by the electron transfer in the PT-ET CO₂ photoreduction. The utilized mythology was well described and fulfill the required standards. However, the conclusions and data discussion needs to be improved. Hence, I would recommend the publication of this study after the followed comments have been addressed.

Q1: *The poor characteristics of the TiO₂ sample low drastically the impact of the study. It is well known that the interaction between the molecule and the semiconductor surface determines the photocatalytic mechanism. In case of TiO₂ the interactions vary depending on the polymorph, exposed facet, concentration of the oxygen vacancies and other defects, the presence of the hydroxyl groups, the Lewis and Bronstedt acidity, energetic*

position of the conduction band and valence band etc. The TiO₂ sample studied here, was not characterized, so the reported mechanism is limited to this particular sample and can not be generalized to other systems.

Response: We are grateful for the insightful comments and meticulous feedback from the reviewer. We agree that the interaction between the molecule and the semiconductor surface is pivotal for the photocatalytic mechanism, and the characteristics of the TiO₂ sample can significantly influence these interactions. In our original manuscript, we used commercially available anatase TiO₂ nanoparticles (20 nm, Shanghai McLean Biochemical Technology Co., Ltd.), and the lack of detailed characterization may limit the generalizability of our conclusions. Taking the reviewer's feedback into account, we have undertaken a comprehensive characterization of our TiO₂ samples and performed additional experiments to validate our findings across different TiO₂ systems.

- 1) **Effect of Crystal Structure:** We examined the influence of the TiO₂ crystal structure by comparing the kinetic isotope effects (KIE) for the CO₂ reduction on anatase and rutile TiO₂ (characterized by XRD and TEM/HRTEM, see **Fig. R1a** and **Fig. R2**). We found that the $K_{SIE_{H_2O/D_2O}}(CO)$ on both anatase and rutile catalysts exhibited inverse values (KIE <1), suggesting that the observed inverse KIE and the protonation mechanism of CO₂ reduction are common to both anatase and rutile crystal structures.
- 2) **Effect of Exposed Facet:** We synthesized anatase TiO₂ nanosheets with a high exposure of the {001} facet (following the methods reported in the literature: J. Am. Chem. Soc., 2017, 139(12): 4486-4492), which were characterized using XRD, TEM, HR-TEM, and SAED (see **Fig. R3**). The $K_{SIE_{H_2O/D_2O}}(CO)$ for CO₂ reduction on these (001)-exposed nanoparticles still exhibited inverse KIE (<1), which is consistent with the protonation pathway. These results further confirmed that the CO₂ reduction pathway under our experimental conditions is not influenced by the exposed facet of the TiO₂ nanoparticles.
- 3) **Effect of Oxygen Defects:** Oxygen vacancies on the TiO₂ surface are often considered active sites for the oxygen evolution reaction (OER). However, their direct influence on CO₂ reduction is less clear. We prepared oxygen-deficient TiO₂ nanoparticles using a reported method (Nanoscale, 2014, 6, 10216–10223), and characterized them using XRD, TEM and ESR, which confirmed the presence of oxygen vacancies (see **Fig. R4**). The KIE for CO₂ reduction on these oxygen-deficient nanoparticles remained <1, exhibiting the secondary inverse KIE and aligning with the protonation pathway.

In summary, according to the reviewer's suggestion, we have performed additional characterizations and experiments, which confirmed that the inverse KIE observed in the CO₂ reduction reaction is intrinsic to the TiO₂ material, regardless of the crystal structure, exposed facet, or oxygen vacancy concentration. Therefore, our findings are relevant and can be generalized to a broader range of TiO₂-based photocatalytic systems. We will incorporate these additional results and discussions in the revised manuscript to improve the clarity and

impact of our study. Please see **Page 5 Line 25-32** and **Page 6 Line 1-19**. We again appreciate the reviewer's insightful and professional comments that have substantially improved the quality and precision of our manuscript.

Fig. R1. (a) XRD patterns of rutile and anatase TiO₂; (b, c) KSIE (CO) values on anatase and rutile TiO₂ catalysts in the H₂O/D₂O systems at different temperatures.

Fig. R2. (a, b, c) TEM/HR-TEM images and SAED pattern of the anatase TiO₂ catalyst; (d, e, f) TEM/HR-TEM images and SAED pattern of the rutile TiO₂ catalyst.

Fig. R3. (a) XRD patterns of the (001) facet exposed TiO_2 nanosheet catalyst and the normal anatase TiO_2 catalyst; (b, c) TEM and HR-TEM images of the (001) exposed TiO_2 nanosheet; (d) KSIE (CO) values on {001} facet exposed TiO_2 nanosheet catalyst in the $\text{H}_2\text{O}/\text{D}_2\text{O}$ systems at different temperatures; (e) SAED pattern of the normal anatase TiO_2 catalyst; (f) SAED pattern of the the (001) facet exposed TiO_2 catalyst.

Fig. R4. (a) XRD patterns of oxygen-deficient TiO₂ and normal anatase TiO₂ catalysts; (b) TEM image of the oxygen-deficient TiO₂ catalyst; (c) EPR spectra of the oxygen-deficient TiO₂ catalyst; (d) KSIE (CO) values of the CO₂ reduction reaction on the oxygen-deficient TiO₂ catalyst in the H₂O/D₂O systems at different temperatures.

Q2: Have the authors tried to perform DRIFTS experiments with labeled C¹³O₂ to rule out the electron transfer step occurring first.

Response: We appreciate the reviewer's comment. In our original manuscript, we utilized H/D isotopically labeled H₂O in our DRIFTS experiments to monitor the evolution of the O=C=O-H⁺/D⁺ signal as the reaction progressed. The primary characteristic vibrational frequency is associated with the C=O stretching vibration, which would be significantly influenced by the substitution of the C isotope.

Therefore, as suggested by the reviewer, we have supplemented our study with additional DRIFTS experiments using C¹³-labeled ¹³CO₂. As depicted in **Fig. R5**, a distinct redshift from 2335 cm⁻¹ to 2293 cm⁻¹ was observed when employing ¹³CO₂, corresponding to the shift of the ¹³C=O stretching vibration signal in

$\text{O}=\text{}^{13}\text{C}=\text{O}\text{-H}^+$ compared to the unlabeled $\text{}^{12}\text{C}=\text{O}$ in $\text{O}=\text{C}=\text{O}\text{-H}^+$ due to the $^{12}\text{C}/^{13}\text{C}$ isotope replacement effect. This finding is highly consistent with our existing experimental results and further validates our conclusions. We have included the additional in-situ DRIFTS results in the revised manuscript (see **Page 9, Line 26-32**). We sincerely appreciate the reviewer's insightful suggestion which has helped strengthen our study.

Fig. R5. In-situ DRIFTS spectra collected at the $\text{TiO}_2/\text{H}_2\text{O}/\text{CO}_2$ (a), $\text{TiO}_2/\text{D}_2\text{O}/\text{CO}_2$ (b), and $\text{TiO}_2/\text{H}_2\text{O}/^{13}\text{CO}_2$ (c) interfaces under constant 365 nm (3W, LED) irradiation in 15 min.

Q3: Control-DRIFTS-experiments of bare TiO_2 , of $\text{TiO}_2/\text{H}_2\text{O}$, $\text{TiO}_2/\text{D}_2\text{O}$, $\text{TiO}_2/\text{H}_2\text{O}/\text{CO}_2$ and $\text{TiO}_2/\text{D}_2\text{O}/\text{CO}_2$ in the dark are missing.

Response: We appreciate the reviewer's comment. As requested, the control DRIFTS-experiments of bare TiO_2 , of $\text{TiO}_2/\text{H}_2\text{O}$, $\text{TiO}_2/\text{D}_2\text{O}$, $\text{TiO}_2/\text{H}_2\text{O}/\text{CO}_2$, $\text{TiO}_2/\text{D}_2\text{O}/\text{CO}_2$ and $\text{TiO}_2/\text{H}_2\text{O}/^{13}\text{CO}_2$ (**Fig. R6**) in the dark were performed and added in the revised manuscript. Please see **Page 7 Line 18-19** and **Supplementary Figure 10**.

Fig. R6. *In-situ* DRIFTS spectra collected at the TiO_2 , $\text{TiO}_2/\text{H}_2\text{O}$, $\text{TiO}_2/\text{D}_2\text{O}$, $\text{TiO}_2/\text{H}_2\text{O}/\text{CO}_2$, $\text{TiO}_2/\text{D}_2\text{O}/\text{CO}_2$ and $\text{TiO}_2/\text{H}_2\text{O}/^{13}\text{CO}_2$ interfaces in the dark within 30 min.

Q4: Page 9. Could the authors please explain this sentence: The decay kinetics of O-H/O-D stretching vibration also exhibited an inverse $\text{KIE}=0.827$ in the presence of CO_2 (Figure 2f), suggesting a changed rate-determining step.

Response: We apologize for any confusion caused by the unclear wording in our manuscript. The sentence "The decay kinetics of O-H/O-D stretching vibration also exhibited an inverse $\text{KIE}=0.827$ in the

presence of CO₂ (Figure 2f), suggesting a changed rate-determining step" was intended to highlight an unusual phenomenon, where the decay of the O-H/O-D signal exhibited a KIE smaller than 1 (0.827) in the presence of CO₂. This finding is inconsistent with the normal KIE (>1) associated with the breaking of an O-H bond in water molecules or dissociated OH⁻, implying that the observed O-H/O-D decay KIE does not solely stem from the direct breaking of the O-H/O-D bond.

In most reactions, the overall rate is determined by the slowest step, known as the rate-determining step. In our system, without CO₂, the direct breakage of the O-H/O-D bond undeniably constitutes the rate-determining step, hence its KIE is greater than 1. However, when CO₂ is present, the slower reduction reaction of CO₂ (in this case, the reduction of the protonated intermediate) becomes the rate-determining step, meaning the direct breakage of the O-H/O-D bond is no longer the rate-determining step. That's why we used the phrase "changed rate-determining step". To make this clearer to the reader, we have rephrased this section and provided more detailed explanations. Please see the revised manuscript (**Page 11, Line 10-15**) for the updated text.

Q5: *Following relevant references are missing: ACS Catal. 2018, 8, 2, 1009–1017, Science 336, 1298 (2012).*

Response: We appreciate the reviewer's suggestion. Corresponding reference was cited in the revised manuscript. Please see **Page 3, Line 8**.

Q6: *Page 6, Ref. 19 is not adequate.*

Response: We appreciate the reviewer for their rigorous attitude. We have revised the corresponding reference in the revised manuscript. Please see **Ref. 31** in the revised manuscript.

Q7: *References are missing: Page 5, ...This temperature-dependent KSIE is consistent with the primary KIE's characteristics for O-H/O-D cleavage during the oxygen evolution reaction (OER), indicating direct O-H cleavage as the rate-determining step of water-splitting; Page 6: ...in line with the theoretical H/D replacement effect.*

Response: We appreciate the reviewer's insightful observation regarding the missing references. In response to this, we have now added the appropriate references to support the statements made on Page 5 and Page 6.

On Page 5, where we discuss the temperature-dependent KSIE being consistent with the characteristics of the primary KIE for O-H/O-D cleavage during the oxygen evolution reaction (OER), indicating direct O-

H cleavage as the rate-determining step of water-splitting, we have added references supporting this concept (see revised manuscript, Page 5, Line 12).

Similarly, on Page 6, where we mention that our observations are in line with the theoretical H/D replacement effect, we have added references that discuss this effect in more detail (see revised manuscript, Page 7, Line 23).

The added references are as follows:

1. Tse E. et al. Observation of an inverse kinetic isotope effect in oxygen evolution electrochemistry. *ACS Catal.* **6**, 5706-5714 (2016).
2. Yang X. et al. Mechanism of Water Splitting and Oxygen–Oxygen Bond Formation by a Mononuclear Ruthenium Complex. *J. Am. Chem. Soc.* **132**, 120-130 (2010).
3. Zhang Y. et al. Pivotal Role and Regulation of Proton Transfer in Water Oxidation on Hematite Photoanodes. *J. Am. Chem. Soc.* **138**, 2705-2711 (2016).
4. Vinyard D. et al. Photosystem II oxygen-evolving complex photoassembly displays an inverse H/D solvent isotope effect under chloride-limiting conditions. *Proc. Natl. Acad. Sci.* **116**, 18917-18922 (2019).
5. Chatterjee S. et al. Concerted Proton–Electron Transfer in Electrocatalytic O₂ Reduction by Iron Porphyrin Complexes: Axial Ligands Tuning H/D Isotope Effect. *Inorg. Chem.* **54**, 2383-2392 (2015).

We thank the reviewer for the valuable suggestion to improve the rigor and completeness of our manuscript.

To Reviewer 3:

Comments:

In their submitted article, Yan et al. attempt to study the kinetic isotope effect for Pt/TiO₂ and TiO₂ in CO₂ reduction and hydrogen evolution. Very different KIEs are observed in the different cases, which is attributed to different reaction intermediates. In particular, the authors propose the formation of an O=C=O-H intermediate as initial step in photocatalytic CO₂ reduction.

The attempt of the authors is well noted, and finding an answer to this intriguing question is of high relevance for the community. However, in the current state of the article, the authors do not achieve this goal. Too many questions are left unanswered, and the hypotheses are not solidly proven. It appears to be too much corrections for a major revision, so the article should be rejected in the current state. However, it might be possible that a completely rewritten version might be submitted again to Nature Communications in the future. In the following I will outline the various issues with the current version of the article.

Q1: *Minor issue: In the introduction, the presented theories on the adsorption of CO₂ and its photocatalytic activation on TiO₂ have also been published (early on) in some noteworthy articles from Europe and the USA.*

It is suggested to cite, in particular, the following articles: (i) M.A. Roberts, H.-J. Freund, Surface Science Reports 25 (1996) 225; (ii) I.A. Shkrob et al. J. Phys. Chem. C 116 (2012) 9461; (iii) A. Pougin et al. Phys. Chem. Chem. Phys. 18 (2016) 10809.

Response: We appreciate the reviewer's insightful suggestion. The mentioned references are indeed seminal works in the field of CO₂ adsorption and photocatalytic activation on TiO₂. We agree that these studies provide important foundations for the theories presented in our manuscript. To ensure our work is contextualized appropriately within the existing body of research, we will include these references in our revised manuscript as **Ref. 12-14**. Again, we are grateful for this valuable feedback.

Q2: *Minor issue: Page 4: The term "polar surface" is not well chosen. Some semiconductors, for example ZnO, have actual polar surfaces, with a dipole moment perpendicular to the surface plane (see here: <https://doi.org/10.1021/acs.chemmater.7b01487>). The crystal structure of TiO₂ does not allow the existence of such polar surfaces by this definition. Instead, the authors might think of a better term, such as "hydrophilic surface" or "hydroxylated surface" or "favoring the adsorption of polar molecules" or other.*

Response: We appreciate the reviewer's expert advice. You are correct that the term "polar surface" is not scientifically accurate when applied to TiO₂, given the structural nature of certain semiconductors like ZnO which exhibit true polar surfaces. Our intention was to convey the propensity of the TiO₂ surface to interact with polar entities, rather than implying it possesses an inherent dipole moment perpendicular to the surface.

Taking your suggestion into account, we agree that a more appropriate term would be "hydrophilic surface," or a surface that "favors the adsorption of polar molecules." These terms more accurately describe the behavior of the TiO₂ surface in the context of our work. Therefore, we have revised the manuscript to replace the term "polar surface" with "hydrophilic surface" as the context dictates. We believe these modifications will improve the clarity and scientific accuracy of our descriptions.

We are grateful for your insightful feedback, which aids us in maintaining the precision and rigor of our research.

Q3: *Minor issue: The raw data of CO production on Pt/TiO₂ in presence of CO₂ should also be reported in Figure S1.*

Response: We appreciate the reviewer's comment. As requested, we have included the raw data of CO production on Pt/TiO₂/CO₂ system in the revised **Supplementary Fig. 1 (Fig. R7)**.

Fig. R7. (a) The H₂ evolution for the water-splitting reaction run in 5 h using the TiO₂ photocatalyst at different temperatures (Pt was loaded as co-catalysts); (b) The CO evolution for the CO₂ reduction reaction run in 5 h using the TiO₂ photocatalyst at different temperatures (without Pt co-catalysts); (c) The CO evolution for the CO₂ reduction reaction run in 5 h using the Pt/TiO₂ photocatalyst at different temperatures (Pt was loaded as co-catalysts).

Q4: Major issue: Some details on the experiments are not clear: Pt was deposited before the start of the hydrogen evolution experiment, but this would imply that the chloride from H₂PtCl₆ is still left in the solution. For the CO₂ reduction tests, it seems that Pt was already present beforehand, but then the chloride would not be present. Could this also be the reason for the very different KSIE observed in the two cases?

Response: We highly appreciate the reviewer's careful observation and thoughtful comment. You're correct in pointing out that the method of Pt addition could indeed affect the progression of the reaction, and we acknowledge that our initial hydrogen evolution experiments did not fully take into account the potential influence of residual chloride ions.

To address this concern, we conducted a supplementary experiment where we pre-loaded Pt onto TiO₂ and thoroughly cleaned it to eliminate any residual chloride ions. This ensured that the conditions were strictly identical to those of the Pt/TiO₂/H₂O+CO₂ system, with the sole difference being the absence of CO₂. As shown in **Fig. R8**, the reaction kinetics were not affected by this change. The kinetic isotope effect for water/deuterium-oxide water (KSIE_{H₂O/D₂O} (H₂)) remained positive and directly dependent on temperature, consistent with our previous results. This effectively rules out the possibility that Cl⁻ ions influenced the reaction kinetics. We again appreciate the insightful feedback that significantly contributes to the clarity and precision of our work.

Fig. R8. KSIE (H₂) values obtained by comparing the H₂ production kinetics of the water-splitting reaction in the H₂O/D₂O systems at different temperatures (Pt was pre-deposited to exclude the influence of chloride ions).

Q5: Major issue: In a related manner as in point 4 above, there is a significant discrepancy between the photocatalytic experiments (in liquid water) and the DRIFTS experiments (with water dosed from the gas phase). Can the water still be considered a solvent in the latter case? Are the reaction conditions sufficiently comparable?

Response: We appreciate your careful review and insightful comment. Indeed, as you rightly pointed out, there exists a significant discrepancy between the photocatalytic experiments conducted in liquid water and the DRIFTS experiments where water is introduced in the gas phase. In the in-situ DRIFTS experiments, water is introduced onto the material surface via a carrier gas, which does not simulate the state of water as a solvent in actual experiments (set-up configuration see **Fig. R9**). This approach, however, was a necessary compromise due to the inherent complexities involved in in-situ infrared testing during CO₂ reduction reactions. The DRIFTS detection beam, which is incident from above, requires an unobstructed catalyst surface for efficient signal detection, thus precluding the presence of liquid coverage on the catalyst. It is not feasible to strictly replicate the conditions of the photocatalytic experiment (i.e., TiO₂ nanoparticles dispersed in water).

Yet, we have attempted the alternative the Attenuated Total Reflectance Fourier Transform Infrared Spectroscopy (ATR-FTIR) set-up to simulate water's solvent state: **Fig. R10a** illustrates the ATR-FTIR setup. In the ATR-FTIR tests, a water film fully covers the catalyst layer, and the gas layer lies above the water layer. This design retains the fluidity of the water solvent, but there is no direct contact between the gas and catalyst. As expected, only negative signals of water decomposition were observed in the in-situ ATR-FTIR spectra (**Fig. R10b**), with no detectable CO₂-related signals.

While direct infrared measurement of the catalyst surface state under aqueous conditions proved unfeasible, the suggestions from our reviewers nevertheless inspired us to conduct the CO₂ reduction KIE experiments under varying water conditions. We no longer used liquid water but instead used water vapor for CO₂ reduction reaction KIE verification. Once again, an inverse KIE appeared (**Fig. R11**), similar to when we used water as a solvent. This verifies that regardless of the form in which water (solvent or vapor) participates in CO₂ reduction, the protonation reaction pathway of CO₂ remains unchanged. More importantly, such a water-vapor induced CO₂ reduction is comparable to our in-situ DRIFTS experiments.

We have added the additional experimental results to the revised manuscript (please see **Supplementary Figure 5**). We greatly appreciate your thorough review, which significantly contributes to enhancing the quality of our paper.

Fig. R9. Schematic diagram of the in-situ DRIFTS set-up.

Fig. R10. (a) Schematic diagram of the ATR-FTIR set-up; (b) *In-situ* ATR-FTIR spectra collected at the TiO₂/H₂O/CO₂ and TiO₂/D₂O/CO₂ interfaces under constant 365 nm (3W, LED) irradiation in 15 min.

Fig. R11. KSIE (CO) values are given by comparing the kinetics of the CO₂ reduction reaction on anatase TiO₂ catalyst with H₂O/D₂O in the vapor state at different temperatures.

Q6: Major issue: Deducing the formation of an O=C=O-H intermediate from the given IR spectra is a very vague hypothesis. It is more likely that the light irradiation changes the charge distribution at the interface, making the adsorption of water less feasible and the CO₂ adsorption more feasible. The species with a

vibrational band at 2335 cm^{-1} might simply be weakly coordinated/bound/adsorbed CO_2 at the surface, which is also likely given the only minor shift compared to gaseous CO_2 ($\sim 2360\text{ cm}^{-1}$). The "band" at 2306 cm^{-1} is a mere guess, since it might alternatively be a dip between two other negative bands. At the same time, there is still a "band" (of similar size as the one assigned by the authors) at or near 2335 cm^{-1} , as before. I admit that DRIFTS is usually a vague technique, and there are not many alternatives (if any) to see adsorbates on real catalysts under reaction conditions. But maybe the authors can make a more clear band assignment if they perform another control experiment where they run the reaction for a certain time under light, then remove the gas phase, and look for stable adsorbates left after reaction. But of course, other methods to prove the suggested surface intermediates are also very welcome.

Response: We greatly appreciate the reviewer's insightful comment and agree that assigning a specific band to a particular intermediate is a challenging task when using surface in-situ DRIFTS characterization. However, distinguishing between FT-IR infrared variations caused by changes in surface adsorption configurations and those resulting from surface photochemical reactions can indeed be accomplished by meticulously controlling experimental conditions as described by the reviewer.

As the reviewer suggested, we carried out a control experiment involved first running the reaction for a specified time under light, followed by the removal of the gas phase using a N_2 flow. By subtracting the equilibrium background in N_2 prior to the experiment, we were able to observe changes in surface-adsorbed species over time. Given that the removal of the CO_2 gas phase would cut off the replenishment of surface CO_2 , a fading CO_2 signal would suggest that the observed signals stemmed from the reaction rather than adsorption. Otherwise, we would observe unchanged, stable adsorbate signals. As illustrated in **Fig. R12**, after the abrupt removal of CO_2 , positive signals of C=O vibration from CO_2 species around 2330 cm^{-1} to 2340 cm^{-1} continued to decrease over time and vanished within tens of seconds. This suggests that the observed CO_2 signals are not from a stable adsorbate but from a surface reaction, consistent with our conclusion.

For more evidence of our assignment of the $\text{O}=\text{C}=\text{O}-\text{H}^+/\text{D}^+$ intermediate, it's important to note that in our experimental system, the in-situ infrared data are not standalone evidence. They corroborate our KIE reaction kinetics experiments. The most direct observation in our system is the inverse kinetic isotope effect ($\text{KIE} < 1$) in CO_2 reduction reactions. This result theoretically can only appear when the rate-determining step, the slowest step, is the backward conversion of the $\text{O}=\text{C}=\text{O}-\text{H}^+/\text{D}^+$ intermediate (**Fig. R13**), i.e., the breaking of the C=O (sp^2) bond to form C-O (sp^3). Therefore, this rate-determining step intermediate should accumulate on the catalyst surface, forming a stable equilibrium. When the light source is turned off, this intermediate gradually disappears from the surface.

The positive peak signal (2335 cm^{-1}) observed with in-situ DRIFTS is consistent with this characteristic,

i.e., it appears during light exposure and disappears after the light is turned off. The 2335 cm^{-1} positive peak infrared characteristic signal is essentially the C=O stretching vibration of the $\text{O}=\text{C}=\text{O}-\text{H}^+$ intermediate, directly affected by the C and O atoms (**Fig. 14a**). We performed a control experiment to verify this assignment. When we replaced CO_2 with C^{13} -labeled $^{13}\text{CO}_2$, the signal moved from 2335 cm^{-1} to 2293 cm^{-1} (**Fig. R14c**). However, if it is only the C=O stretching vibration in CO_2 , the C=O stretching vibration frequency should not change other than the direct influence from the C and O atoms. Yet, when we replaced H_2O with D_2O , the signal of the C=O stretching vibration moved from 2335 cm^{-1} to 2306 cm^{-1} (**Fig. 14b**). This shift suggests that the proton directly affects the observed C=O stretching vibration frequency, and its influence is close to the effect of replacing ^{12}C with ^{13}C . This can only mean that the H^+/D^+ adhesion to the O atom in $\text{C}=\text{O}-\text{H}^+/\text{D}^+$ increases the effective mass of the O atom, thereby reducing the frequency of the C=O stretching vibration according to Hooke's law. We conducted mathematical verifications of the influence of H^+/D^+ adhesion on the C=O stretching frequency on the O atom, the results of which are shown in eqs 1-2. This is our most significant basis for assigning the $\text{O}=\text{C}=\text{O}-\text{H}^+/\text{D}^+$ intermediate. Besides, the bending vibration of $\text{C}=\text{O}-\text{H}^+$ at 1089 cm^{-1} was also shifted to 1054 cm^{-1} in the $^{13}\text{CO}_2$ system corresponding to $^{13}\text{C}=\text{O}-\text{H}^+$, where the $\text{C}=\text{O}-\text{D}^+$ bending vibration likely shifted beyond our in-situ DRIFTS detection range. These findings are highly consistent with our KIE experimental results and further validates our assignment. All these assignments were backed by quantum chemical calculations, which will be discussed below in the response of Q7.

In summary, the additional experimental result and clearer discussions on the assignment of the $\text{O}=\text{C}=\text{O}-\text{H}^+$ intermediate have been added to the revised manuscript (please see **Page 9 Line 4-14, Line 20-22, Line 25-32, and Supplementary Figure 11**). We greatly appreciate your thorough review, which significantly contributes to enhancing the precision of our paper.

Fig. R12. (a) In-situ DRIFTS spectra taking with the bare TiO_2 surface (in N_2 flow) as the background then CO_2 was introduced into the system in the dark for 15 min. (b) After 15 min in the dark, the identical system

of (a) was then illuminated by a 365 nm (3W) LED lamp for 10 min. (c) during the continuous 365 nm (3W, LED) irradiation, an abrupt removal of the CO₂ gas-phase by the N₂ flow was conducted. (d) Waterfall profiles of the in-situ DRIFTS spectra after the abrupt CO₂ gas-phase removal. Surface CO₂ species faded out in 1 min.

Fig. R13. Schematic illustrations and energetic profiles of the $\text{O}=\text{C}=\text{O}-\text{H}^+/\text{D}^+ \rightarrow \text{O}=\text{C}^{\cdot}-\text{O}-\text{H}/\text{D}$ electron transfer process.

Fig. R14. In-situ DRIFTS spectra collected at the TiO₂/H₂O/CO₂ (a), TiO₂/D₂O/CO₂ (b), and TiO₂/H₂O/¹³CO₂ (c) interfaces under constant 365 nm (3W, LED) irradiation in 15 min.

Q7: Major issue: In continuation of comment 6, I would also suggest quantum chemical calculations of the vibrations of the O=C=O-H intermediate. The strong band of (gaseous or weakly adsorbed) CO₂ at/near 2360 cm⁻¹ is the asymmetric O=C=O stretching frequency. But in CO₂, both O atoms are identical, giving rise to only one band. A species in which one oxygen atom binds or coordinates a hydrogen atom would resemble more a carboxyl species, which features very different bands for the two different O=C and C-O(-H) bonds. I am doubtful that such a species would have one vibrational band near 2335 cm⁻¹.

Response: We appreciate the reviewer's insightful comment. Indeed, as noted, the observed negative and positive signals in the 2300~2350 cm⁻¹ band derive from the O=C=O stretching vibration. The two negative peaks likely correspond to strongly and weakly adsorbed (or gaseous) CO₂, while we attribute the positive peak to the O=C=O-H⁺ intermediate. It's important to note that the C=O bond in the C=O-H⁺ stretching vibration remains sp² hybridized. The proton merely adheres to the oxygen atom through protonation, and electron transfer has not yet occurred. Therefore, its vibration mode is only affected by altered bond energy and changes in effective mass, and it does not directly form a sp³ hybridized hydroxyl-like pattern as in carboxyl C-O-H, as this requires subsequent electron transfer, as shown in Fig. R15.

Following the reviewer's suggestion, we conducted related quantum chemical calculations to study the infrared signals of the H⁺/D⁺ protons adhered to the oxygen atom in CO₂ (Fig. R16). The results are consistent with our assumption that the C=O stretching vibration in CO₂ does not form a C-O-H sp³ structure after adhering to a H⁺/D⁺ proton, thereby a C-O signal does not appear. Instead, it remains in the range of 2300-2400 cm⁻¹. The vibration frequency discrepancy between protonated species and pristine CO₂ is due to the influence of bond energy and the effective mass of oxygen.

Moreover, replacing H⁺ with D⁺ indeed causes the simulated C=O stretching vibration to shift to a lower frequency (2403 cm⁻¹ → 2394 cm⁻¹). Interestingly, quantum calculations also reveal possible O-H/O-D stretching vibrations (3406 cm⁻¹/2490 cm⁻¹), which are not clearly observed in the experiment due to the significant influence of water signals. More importantly, we found that the 960 cm⁻¹ in O=C=O-H⁺ corresponds to the bending vibration of C=O-H⁺, which might correspond to the positive signal at 1089 cm⁻¹ observed in in-situ DRIFTS. In O=C=O-D⁺, the bending vibration of C=O-D⁺ shifts to a lower frequency, beyond our in-situ DRIFTS detection range, making it difficult to observe. However, when ¹³C is used for simulation, the bending vibration of ¹³C=O-H⁺ can be seen to shift from 960 cm⁻¹ to 952 cm⁻¹. In our actual in-situ DRIFTS experiments, when using ¹³CO₂, we indeed observed a shift towards a lower wavenumber of the ¹³C=O-H⁺ bending vibration (1054 cm⁻¹) from C=O-H⁺ (1089 cm⁻¹) using unlabeled CO₂ (Fig. R17). This further provides additional support for our assignment.

In summary, our quantum chemical calculations, combined with our experimental results, provide additional support for our assignment of the observed vibrational bands. All additional experimental and simulation results and corresponding discussions have been added to the revised manuscript (Please see **Page 10, Line 1-17**; and **Supplementary Figure 12**). We appreciate the reviewer's suggestion to perform these calculations, which have significantly enhanced our understanding and interpretation of the results.

Fig. R15. Schematic illustrations and energetic profiles of the $\text{O}=\text{C}=\text{O}-\text{H}^+/\text{D}^+ \rightarrow \text{O}=\text{C}-\text{O}-\text{H}/\text{D}$ electron transfer process.

Fig. R16. (a) Quantum chemical calculation of the infrared spectra of CO_2 , $\text{O}=\text{C}=\text{O}-\text{H}^+$ and $\text{O}=\text{C}=\text{O}-\text{D}^+$; (b) Quantum chemical calculation of the infrared spectra of $^{13}\text{CO}_2$, $\text{O}=\text{}^{13}\text{C}=\text{O}-\text{H}^+$ and $\text{O}=\text{}^{13}\text{C}=\text{O}-\text{D}^+$.

Table 1 Assignment of CO_2 , $\text{O}=\text{C}=\text{O}-\text{H}^+/\text{D}^+$, $\text{O}=\text{}^{13}\text{C}=\text{O}-\text{H}^+/\text{D}^+$ infrared spectral vibrational modes by quantum

chemical calculations.

	Wavelength (cm ⁻¹)	Vibration mode
CO ₂	642	O=C=O bending vibration
	2349	O=C=O asymmetric stretching vibration
O=C=O-H ⁺	960	C=O-H bending vibration
	2403	O=C=O asymmetric stretching vibration, C-H stretching vibration
	3406	O-H stretching vibration
O=C=O-D ⁺	469	O=C=O bending vibration, C-D bending vibration
	561	O=C=O bending vibration
	784	C=O-D bending vibration
	1214	O=C=O symmetric stretching vibration
	2394	O=C=O asymmetric stretching vibration, C-D stretching vibration
	2490	O-D stretching vibration
¹³ CO ₂	624	O=C=O bending vibration
	2282	O=C=O asymmetric stretching vibration
O= ¹³ C=O-H ⁺	952	C=O-H bending vibration
	2337	O=C=O asymmetric stretching vibration, C-H stretching vibration
	3406	O-H stretching vibration
O= ¹³ C=O-D ⁺	461	O=C=O bending vibration, C-D bending vibration
	545	O=C=O bending vibration
	775	C=O-D bending vibration
	1212	O=C=O symmetric stretching vibration
	2329	O=C=O asymmetric stretching vibration, C-D stretching vibration
	2487	O-D stretching vibration

Fig. R17. *In-situ* DRIFTS spectra collected at the $\text{TiO}_2/\text{H}_2\text{O}/\text{CO}_2$ (a), $\text{TiO}_2/\text{D}_2\text{O}/\text{CO}_2$ (b), and $\text{TiO}_2/\text{H}_2\text{O}/^{13}\text{CO}_2$ (c) interfaces under constant 365 nm (3W, LED) irradiation in 15 min.

Q8: *Minor issue: Page 7: "Hooker's law" should be "Hooke's law".*

Response: We appreciate the reviewer's comment. Corresponding error has been corrected.

Q9: *Major issue: In the experimental section, the authors do not report any particular cleaning of the TiO_2 before the reaction. Formation of carbon-containing "products" from impurities is a known observation for TiO_2 (see works by Guido Mul, Elena Selli, Adriana Zaleska and Jennifer Strunk from ~2010 to 2019). Since the authors perform water splitting before adding CO_2 , which can reasonably be considered a blank experiment for CO_2 reduction (see, e.g., Moustakas and Strunk, *Chem. Eur. J.* 24, 2018) such formation of byproducts should be observed. Did the authors really observe "no" CO (or CH_4) during the initial water splitting experiments? Or in other words, can the formation of CO from sources other than CO_2 be clearly excluded?*

Response: We appreciate the reviewer's comment. Indeed, our TiO_2 catalyst undergoes high-temperature calcination prior to use and is pre-irradiated with UV light (365nm) in water to remove possible impurities. However, as the reviewer rightly points out, the formation of carbon-containing products such as formic acid,

CO, or CH₄ from impurities is a well-known occurrence, and it is nearly impossible to completely exclude it, even with high-temperature calcination or other purification methods. However, the amount of these impurity-derived products compared to the products of CO₂ photoreduction is negligible.

We compared the CO production in systems with and without CO₂ (**Fig. R18a**). The results showed that the CO production in the system without CO₂ is negligible compared to the system with CO₂. This discrepancy does not affect our observed KIE tests. To further verify the purity of the CO product, we conducted isotope labeling mass spectrometry experiments using ¹³C-labeled ¹³CO₂. The content of ¹³CO overwhelmingly dominated, indicating that the reduction of CO₂ is undoubtedly the main source of CO (**Fig. R18b**).

In summary, while we acknowledge that the formation of carbon-containing products from impurities cannot be completely excluded, our experimental evidence convincingly demonstrates that the majority of the observed CO originates from CO₂ reduction. The description of catalyst cleaning and additional control experimental results were added to the revised manuscript, please see **Page 5 Line 3-5**. We again appreciate the reviewer's suggestion, which prompted us to further validate our findings.

Fig. R18. (a) The comparison of CO production rate with/without CO₂ on anatase TiO₂ catalyst (b) Mass spectrometry analyses of ¹³CO (m/z =29) when using ¹³CO₂ in the CO₂ photoreduction on the TiO₂ catalyst (As follows: the raw data report with mass spectrometry).

REVIEWERS' COMMENTS

Reviewer #2 (Remarks to the Author):

The authors have addressed the comments made by the reviewers in the revised version of the manuscript. Therefore, I have no further comments and the manuscript can be published as it is.

Reviewer #3 (Remarks to the Author):

In the revised version, the authors invested a lot of time and effort to clarify all open questions. The quantum chemical calculations and the isotope labelling studies are particularly positively noted. I checked the whole manuscript carefully again, and I think that now all hypotheses are well supported by the data.

Reply to reviewers' comments

To Reviewer 2:

Comments:

The authors have addressed the comments made by the reviewers in the revised version of the manuscript. Therefore, I have no further comments and the manuscript can be published as it is.

Response: We appreciate the reviewer's effort in reviewing our work.

To Reviewer 3:

Comments:

In the revised version, the authors invested a lot of time and effort to clarify all open questions. The quantum chemical calculations and the isotope labelling studies are particularly positively noted. I checked the whole manuscript carefully again, and I think that now all hypotheses are well supported by the data.

Response: We appreciate the reviewer's effort in reviewing our work.